# Goal-oriented possibilistic fuzzy C-Medoid clustering of human mobility patterns— Illustrative application for the Taxicab trips-based enrichment of public transport services

**Miklós Mezei[1], Imre Felde[2], György Eigner[2,3]\*, Gyula Dörgő[4], Tamás Ruppert[4], János Abonyi[4]**

**1** Kálmán Kandó Faculty of Electrical Engineering, Department of Automation, University of Óbuda, Budapest, Hungary, **2** John von Neumann Faculty of Informatics, Biomatics and Applied Artificial Institution, Óbuda University, Budapest, Hungary, **3** Physiological Controls Research Center, Research and Innovation Centre, Óbuda University, Budapest, Hungary, **4** MTA-PE Lendület Complex Systems Monitoring Research Group, Department of Process Engineering, University of Pannonia, Veszprém, Hungary

\* eigner.gyorgy@nik.uni-obuda.hu

**Data Availability Statement:** Data requests can be made via: titkarsag@nik.uni-obuda.hu

**Funding:** Project no. 2019-1.3.1-KK-2019-00007. has been implemented with the support provided

## Abstract

The discovery of human mobility patterns of cities provides invaluable information for decision-makers who are responsible for redesign of community spaces, traffic, and public transportation systems and building more sustainable cities. The present article proposes a possibilistic fuzzy c-medoid clustering algorithm to study human mobility. The proposed medoid-based clustering approach groups the typical mobility patterns within walking distance to the stations of the public transportation system. The departure times of the clustered trips are also taken into account to obtain recommendations for the scheduling of the designed public transportation lines. The effectiveness of the proposed methodology is revealed in an illustrative case study based on the analysis of the GPS data of Taxicabs recorded during nights over a one-year-long period in Budapest.

## Introduction

According to the UN reports, cities are responsible for approximately 70% of global carbon emissions, and the expected population living in cities will reach 6.5 billion by 2050 [1]. The transport sector is one of the main contributors to greenhouse gas emission. Rapid urban population growth, traffic congestion, and related air pollution put cities at the center of the climate mitigation agenda. These facts suggest urgent and transformative actions in urban mobility are required [2]. According to the report of Masson-Delmotte *et al.* on global warming [3], in 2014, transportation accounted for 23% of global energy-related $CO_2$ emissions and by 2017 the impact of road transport was further increased by 2%, from which 44% was caused by passenger cars [4]. Another report from the European Commission states that, urban mobility accounts for 40% of all $CO_2$ emissions of road transportation and up to 70% of other pollutants from transportation [5]. In slight contrast, another study by Toledo *et al.* found that

from the National Research, Development and Innovation Fund of Hungary, financed under the 2019-1.3.1-KK funding scheme. Funder: National Research, Development and Innovation Office https://nkfih.gov.hu/for-the-applicants The funders did not play any role regarding the study.

**Competing interests:** NO authors have competing interests.

individual motorized transport causes 59% of greenhouse gas emissions [6]. Shapiro *et al.* compared the emissions of private vehicles and public transportation, and found that public transport produces 95% less CO, 45% less $CO_2$, and 48% less $NO_2$ than private vehicles [7].

According to the work of Jenks *et al.* on the dimensions of sustainable cities, the sustainability of cities depends on environmental, transportation, social and economic issues [8]. This is well complemented by the nowadays trending smart-city concept, which supports the different fields of urban mobility to decrease carbon emissions [2]. Smart-mobility applications like mobile monitoring systems [9], traffic performance measurement [10], bicycle-sharing systems [11] and smart vehicle routing systems [12] were proved to have an advantageous environmental impact in facilitating air-pollution reduction. Based on the findings of Jenks *et al.*, the improvement of transportation should be based on a better understanding of the impact of the urbanization form on travel behaviours [8]. By optimizing public transportation based on human mobility patterns, there is a possibility to avoid constant traffic jams and decrease pollution. The Taxicab trips provide a valuable and unique source of information to explore human mobility patterns of the city.

As it was pointed out by Siła-Nowicka *et al.*, the analysis of human mobility patterns is essential for understanding the evolution of size and structure of urban areas [13]. The primary goal of the analysis of these mobility patterns is to get a better overview of the system design. This trend of analysis has gained momentum, as the Internet of Things (IoT) equipment that captures movement information in real-time and at detailed spatial and temporal scales (*e.g.*, GPS trackers [14]) has changed the ability to collect movement data [15]. These GPS trajectories enable the exploration of formerly hidden aspects of the dynamics of cities. Cities have introduced the concept of GPS sensor-equipped Taxicabs to enable Taxicab tracking services, which generate GPS trace mobility data [16]. As it was highlighted by Kumar *et al.*, this position information provides helpful insight into the human mobility patterns of the city [17]. As an example, the work of Kaltenbrunner *et al.* illustrates how the data recorded by the bicycle sharing system was utilized to detect temporal and geographic mobility patterns within the city [18]. Böhm *et al.* used GPS traces and a microscopic model to analyse the emissions of four air pollutants from thousands of vehicles in three European cities [19]. Chen *et al.* presented how these trajectories could be used to analyse the emission of particle matter from braking behaviours [20]. Human mobility analysis approaches were overviewed, and two predictions (next-location- and crowd flow prediction) and two productive tasks (trajectory- and flow generation) were discussed in the work of Massimiliano *et al.* [14]. Mobility pattern mining is also used to understand the group-based travel behaviours as presented by Du *et al.* [21]. The analysis helps to diagnose and understand the residence of each region with their demand for public transportation. This is especially crucial, as according to Egger [22], transportation choices are a fundamental component of the sustainability due to their relevant impact on the economy and the further social, political, and environmental aspects. According to Badia *et al.*, the convenience of transit systems versus cars in urban areas is generally well-accepted [23], and in particular, electric bus-based public transportation systems should be designed to improve the sustainability of the cities according to the work of Majumder *et al.* [24]. For the design of these networks, a multi-stage machine learning framework has been developed in the work of Tang *et al.* to predict boarding stops of passengers based on recurrent neural networks (RNN) [25]. Based on data-driven models, the stops of bus trips can also be estimated for public transport planning as it was presented in another work of Tang *et al.* [26]. The bus driving cycles were also analysed, where the on-road bus speed data were extracted from GPS data, identifying five significantly different bus-driving patterns [27]. In another work by AlRukaibi and AlKheder, the bus stopping stations are optimized in Kuwait, where a standard distance is proposed to keep 1–1.4 km between every two stops [28].

As it was described by Kumar *et al.*, Taxicabs have comprehensively good coverage of the city, hence provide a basis for a reasonably good estimation of general mobility trends of people and city hotspots [17]. In their work, the trajectories of Taxicab positions are represented by the sequence of GPS points or the origin-destination pair for each passenger ride. The clustering of origin-destination locations provides valuable insight into the passenger movement and helps to identify where the Taxicab drivers are most likely to find their next customer [17]. Clustering is also an efficient approach to get an adaptive routing method for the cruising Taxicabs by suggesting vacant Taxicabs to the pathways having many potential passengers as showed in the work of Yamamoto *et al.* [29]. Data mining techniques, such as clustering and naive Bayesian classifier, are also applicable to historical data for building models and predicting Taxicab demand in context of time, weather, and location [30, 31]. The mobility patterns within the city of Singapore were analysed in the work of Kumar *et al.* by density-based clustering of origin-destination pairs of the passenger Taxicab rides using the DBSCAN algorithm [17]. Density-based hierarchical clustering method (DBH-CLUS) is used to identify pick-up/drop-off hotspots by Wan *et al.* [32], and the spatio-temporal patterns in the passenger movements are discovered using spatial clustering of the origin-destination data pairs in the work of Guo *et al.* [33].

Although, clustering is an efficient approach for the grouping of the rides and detecting relevant and frequent mobility patterns, its application for the design of public transportation lines reveals three major deficiencies and practical problems/aspects:

- The outliers shift the cluster centroids, significantly hindering the detection of relevant patterns.

- As we would like to avoid the ad-hoc installation of new public transportation stops, the cluster centroids should be selected from the existing stops of the city.

- The assignment of rides to public transportation stops is not arbitrary, only rides starting within a walkable distance should be considered as the member of a cluster.

The k-means algorithm is capable to solve the practical segmentation problems [34, 35], while the classical Fuzzy C-means (FCM) [36] approach is the better choice for spherical clusters [37]. The classical FCM uses a variant of distance-based measure to define the distance between the cluster center and members. The Possibilistic Fuzzy c-means (PFCM) algorithm is introduced by Pal *et al.* [38] to reduce the effect of outliers in a cluster by the introduction of a typicality factor in the cost function. This algorithm was further modified by Király *et al.* [39] to retrieve the cluster centroids from a pre-defined set and form a Fuzzy C-medoid solution. The possibilistic approach to clustering aims to address the problems associated with the constraint on the membership used in FCM. Foremost, the main difference between FCM and Possibilistic C-means (PCM) [40] is in the membership representation. In the fuzzy case, each point is the member of different clusters at a particular ratio (the sum of the membership values of each point is 1), so the constraint used by the FCM approach can be interpreted as a shared degree of membership value (What is the ratio of the specific point in the cluster membership?) but not as degrees of typicality (How typical is the specific point in the cluster?) [41]. The membership value in a cluster represents the possibility of the point belonging to the cluster. On the other hand, the typicality of the point in the cluster features how typical the point in the specific cluster is. Since noise points or outliers are less typical in a cluster, typicality-based memberships automatically reduce the effect of noise points and outliers, and considerably improve the results.

The daytime bus transportation schedules in many cities are usually well designed [42]. Late at night, Taxicab is the only way for getting around. Formerly, the night-bus route planning

problem is investigated by leveraging Taxicab GPS traces based on the expected number of passengers along the routes [42]. Similarly, the daytime public transportation in the city of Budapest is relatively dense, hence we focused on the analysis of the late-night Taxicab rides to 1) Identify the mobility of the city 2) Make recommendations for the design of public transportation lines. The developed PFCMD clustering algorithm aims to cluster the start and end positions of Taxicab rides to public transportation stops to see whether a well-organized public transport line could replace the group of these Taxicab rides. The resultant rides are grouped according to their position, while the start time of the lines can be determined by the temporal analysis of the start times of Taxicab rides in the specific group. The frequently occurring start times indicate when the lines obtain the most significant possibility of replacing individual Taxicab rides. The developed analyses can also help to optimize the efficiency of the Taxicab service.

We aim at modifying the PFCM clustering algorithm to Possibilistic Fuzzy C-medoid (PFCMD) to find the clusters based on the pre-defined set of possible central points and group the taxi rides within walking distance to these centroids. On the grounds of the aforesaid, the contribution of the present paper is to fully describe the developed novel Possibilistic Fuzzy C-medoid (PFCMD) clustering algorithm and prove its applicability for the discovery of human mobility patterns based on public transportation schedules during the night shifts at Budapest.

The roadmap of the paper is as follows. The developed PFCMD algorithm is described with the problem formulation in the Method section, this is where the methods of temporal analysis are also detailed. The analysed dataset that contains the nightly taxi rides in Budapest over a year-long period, the effect of clustering parameters and the comparison of the clusters identified by the PFCMD algorithm and the k-medoid-based solutions are showcased in the Results section. Finally, the results are discussed, and the article is concluded with some last remarks in the Conclusions section.

## Methods

In this section, the developed PFCMD clustering algorithm is defined. Firstly, we introduce the problem formulation. After that, the detailed description of the algorithm follows, and finally, the temporal analysis is briefly profiled.

Let $\mathbf{R} = [\mathbf{r}_1, \mathbf{r}_2, \ldots, \mathbf{r}_N]$ be a given set of $N$ patterns, $n = 1 \ldots N$, each of them representing a mobility pattern as a Taxicab's ride. Therefore, the $n^{th}$ pattern is defined by $\mathbf{r}_n = (\mathbf{p}_{ns}, \mathbf{p}_{ne}, t_{ns}, t_{ne})$, where $\mathbf{p}_{ns} = [p_{ns1}, p_{ns2}]$ denotes the start (pickup) and $\mathbf{p}_{ne} = [p_{ne1}, p_{ne2}]$ indicates the end (drop-off) GPS latitude ($p_{ns1}$ and $p_{ne1}$) and longitude ($p_{ns2}$ and $p_{ne2}$) coordinates, and $t_{ns}$ and $t_{ne}$ are the start and end times, respectively.

The Taxicab trips are defined based on the state identifier of the Taxicab, indicating the operation mode of the Taxicab. Therefore, pickups are recorded when the state identifier is changed from *Free* (0) to *Occupied* (1), while the drop-off is indicated by the change of the state identifier from *Occupied* (1) to *Free* (0). Moreover, we have a Taxicab identifier, but the workload of different Taxicabs was not analysed. The stations of public transportation are determined by $\mathbf{s}_j \in \mathbf{S}$, $j = 1 \ldots N_s$ stations where $\mathbf{s}_j = [s_{j1}, s_{j2}]^T$ denotes the GPS latitude and longitude coordinates of the stations. We aim to assign the Taxicab rides to these stations based on the pickup and drop-off coordinates, and find a reasonable schedule for these lines. The grouping of the rides to public transportation stations is performed by clustering, while the design of the line schedule is defined by the time series analysis of the grouped rides.

### The goal-oriented Possibilistic Fuzzy C-medoid algorithm (PFCMD)

As the clustering is performed in the geographical domain, only pickup and drop-off coordinates are used in this step of the methodology and for easier notation, the $\mathbf{x}_n$ is reduced to a

vector containing the coordinate-based records $\mathbf{x}_n = [\mathbf{p}_{ns}, \mathbf{p}_{ne}]$, therefore, clustering is realized in a four dimensional space: $\mathbf{x}_n = [p_{ns1}, p_{ns2}, p_{ne1}, p_{ne2}]$. These points are to be partitioned into $C$ clusters. The prototype of the $c^{th}$ cluster is denoted by $\mathbf{v}_c = [\mathbf{s}_i, \mathbf{s}_j]$, where $\mathbf{s}_i, \mathbf{s}_j \in \mathbf{S}$ and $i \neq j$. The original PFCM algorithm [38] aims to minimize the following optimization problem:

$$J_{m,\eta}(\mathbf{U}, \theta, \mathbf{V}; \mathbf{X}) = \sum_{n=1}^{N}\sum_{c=1}^{C}(au_{cn}^m + b\tau_{cn}^\eta) \times ||\mathbf{x}_n - \mathbf{v}_c||^2 + \sum_{c=1}^{C}\gamma_i\sum_{n=1}^{N}(1 - \tau_{cn})^\eta \qquad (1)$$

subject to constraints $\sum_{c=1}^{C} u_{cn} = 1 \forall n$, and $0 \leq u_{cn}, \tau_{cn} \leq 1$, while $m \geq 1, \eta \geq 1, \gamma_i > 0$. $\mathbf{u}_c$ represents the $c^{th}$ row of the membership matrix ($\mathbf{U}$) and contains all the memberships associated with the $c^{th}$ cluster. The typicality is represented by the typicality matrix $\theta = [\tau_{cn}]_{C \times N}$, the $\mathbf{V} = [\mathbf{v}_1, \ldots, \mathbf{v}_C]$ is the matrix of cluster centres and $\mathbf{X}$ is the analysed dataset. The user defined constants are the relative importance of fuzzy membership $a > 0$ and the typicality value $b > 0$. The membership value, $u_{cn}$, of a point in a cluster represents the membership of $\mathbf{x}_n$ in the $c^{th}$ cluster. Originally, in fuzzy c-means clustering [43], the membership values of a data point are inversely proportional to the relative distance of the data point to the $C$ cluster prototypes. However, assuming $C = 2$ and an equidistant data point from the two cluster centroids, the membership value of the data point in each cluster is 0.5, regardless of the absolute distance of the data point to the cluster centroids. Therefore, noise points far but equidistant from the cluster centroids would produce equal membership values in both clusters, instead of the more natural choice of very low cluster membership values. To overcome this problem, the typicality of a point in a cluster was introduced, $\tau_{cn}$, which is interpreted as how relatively typical the point in cluster $C$ is [40]. Therefore, taking advantage of both approaches, Pal *et al.* combined these terms into a single cost function [38].

If $D_{cn} = ||\mathbf{x}_n - \mathbf{v}_c|| > 0$ for all $C$, where the $||\mathbf{x}_n - \mathbf{v}_c||$ notation describes a standard L2 vector norm, then the membership and typicality values are calculated based on Eqs 2 and 3, respectively.

$$u_{cn} = \left(\sum_{j=1}^{C}\left(\frac{D_{cn}}{D_{jn}}\right)^{2/(m-1)}\right)^{-1} \qquad (2)$$

In the present work, we change the original typicality function of Pal *et al.* [38] for a flexible negative Gompertz function of the distance as presented in Eq 3, which models the willingness of people to walk between, to and from the nearest public transportation stop instead of choosing a door-to-door transportation method.

$$\tau_{cn} = 1 - \alpha e^{-\beta e^{-\gamma D_{cn}}} \qquad (3)$$

The $\alpha, \beta$ and $\gamma$ are the parameters of the typicality function, making it highly flexible for the definition of a desirability trend. In the present context, this means the connection of rides being close to the public transportation stop.

The possible centroids are selected from a predefined set of points, in the present context the public transport stops.

$$v_{c1} = \arg\min_i \sum_{n=1}^{N}(au_{cn}^m + b\tau_{cn}^\eta)D([p_{ns1}, p_{ns2}]^T, \mathbf{s}_i)^2 \qquad (4)$$

$$v_{c2} = \arg\min_i \sum_{n=1}^{N}(au_{cn}^m + b\tau_{cn}^\eta)D([p_{ne1}, p_{ne2}]^T, \mathbf{s}_j)^2, \qquad (5)$$

where $1 \leq c \leq C$; $1 \leq n \leq N$ and $D([p_{ne1}, p_{ne2}]^T, \mathbf{s}_j)^2$ represents the distance between the data-point (starting or ending of the ride) and the public transport stop that represents the center of the given cluster.

Finally, as the cluster centroids, membership and typicality values are determined, the $\mathbf{x}_n$ data point is considered to be the member of each cluster, where the combined cluster membership value is above a certain user-defined threshold, $P_{threshold}$:

$$au_{cn}^m + b\tau_{cn}^\eta > P_{threshold} \tag{6}$$

We applied the Partition Coefficient (PC) and the Classification Entropy (CE) to evaluate the quality of the clusterings:

$$PC = \frac{1}{N}\sum_{c=1}^{C}\sum_{n=1}^{N}(u_{cn}^m)^2 \tag{7}$$

$$CE = -\frac{1}{N}\sum_{c=1}^{C}\sum_{n=1}^{N}u_{cn}^m log(u_{cn}^m) , \tag{8}$$

where CE values close to zero and PC values close to one indicate well-separated cluster structure [44].

## Results

Human mobility patterns analysis proves the applicability of the proposed PFCMD algorithm in the Hungary capital city, Budapest. We focused on the night shifts to compare the most frequent patterns with the possible public transportation stops based on the C-medoid clustering method and discover the possible public transportation routes. In this section, first, the analysed dataset, the Taxicab rides data recorded during the nights in Budapest are introduced. This is followed by the discussion of the proposed clustering-based solution, paying special attention to parameter tuning. Finally, the recommendation for the schedule of the possible public transportation lines is proposed by the temporal analysis of the start time of the rides.

### The analysed taxicab rides of Budapest

The proposed PFCMD algorithm is applied to location data from Taxicabs equipped with a GPS receiver and an interface to record the actual state of the Taxicabs (engaged, vacant, not in service or en route for an incoming carriage request) [16]. The analysed GPS data was recorded in 2014 in Budapest and contained 450 million position records of 801 different city Taxicabs. The public transportation data comes from the official Budapest public transportation company (BKK Budapesti Közlekedési Központ Zrt.). The dataset contains all information about the BKK lines incorporating the routes, stops, stop times, and trip information in standard General Transit Feed Specification (GTFS) [45] format. From this available information, our analysis utilizes the coordinates of the public transportation stops. As the public transportation system of the city can be considered quite dense both spatially and temporally, in our work, we focused on the night rides with the starting time beginning after 9:00 PM and ending before 6:00 AM. Fig 1 illustrates the relatively dense and well-distributed public transport network of Budapest, which is overviewed with the Taxicab routes. Therefore, our research question is whether Taxicab rides can be more sustainably replaced by well-planned public transport solutions (mainly buses). Are there significant hubs that should be connected? Are there frequent times that should be better served at nights?

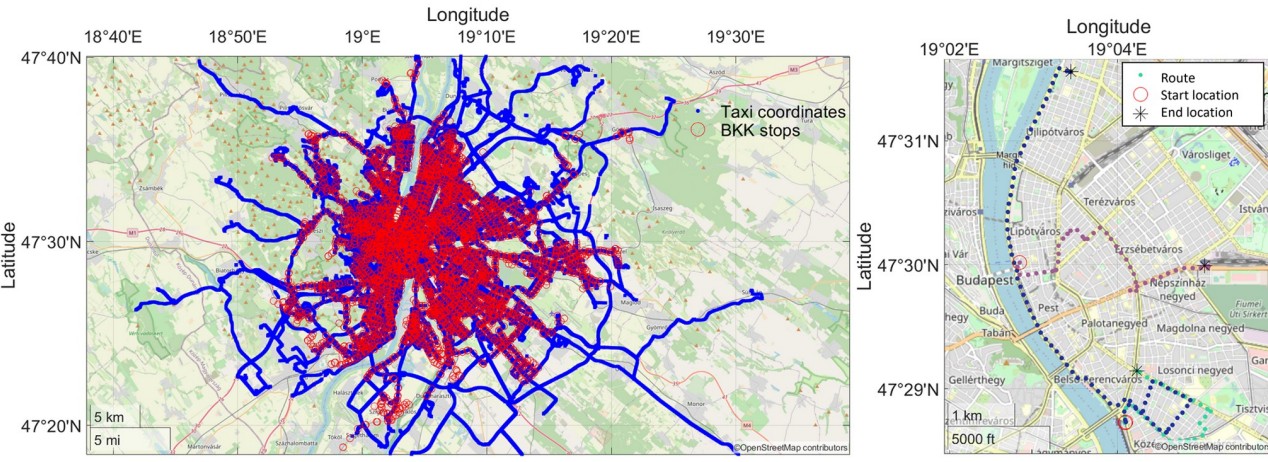

**Fig 1. Stop stations of public transportation (red circles) and the travels of the Taxicabs (black lines) on the left.** Some example rides with start and end location are plotted on the right side of the figure.

The resultant 436537 rides during the analysed nights of 2014 mean an average of $\sim$ 1196 rides per night. We can assume that the barrier of changing from Taxicabs to public transportation is high for some passengers or in some cases. Moreover, the topology of Budapest can be considered quite complex, as the city is divided by the river Danube and the nightlife is mainly concentrated on the eastern side, leaving the western side calmer and less dense. In this regard, it is apparent that the planning of public transportation in Budapest is a highly complex challenge, and a careful analysis of the rides is required to ensure the utilization of the designed lines by the passengers.

The existing public transportation lines are not included in the current analysis. However, the results are comparable with the existing routes, and decision-makers can make recommendations to re-route existing lines or introduce new ones.

## Clustering the public transportation data

Our main questions were: Are there significant hubs that should be connected? Are there times that should be better served during nights? By detecting the major mobility patterns of Taxicab rides and comparing the designed lines to the existing public transportation system, previously uncovered areas can be connected by introducing new lines. In order to detect the start- and end-points of these lines, we clustered the Taxicab rides using the developed PFCMD algorithm with the previously defined parameter setting and initialized from the results of a k-medoid clustering.

The advantages of the applied algorithm are visible in Fig 2. The result of k-medoid clustering consists of several outlier clusters, which are indicated by the conspicuous red lines. It is clear that the public transportation system cannot aim to cover these occasional rides sometimes pointing out of the city, but instead it should strive to meet the needs of the bulk of the community. A very striking example is the trip to Vienna (the long red line in the left part of Fig 2), which was covered by the traditional k-medoid clustering solution and a separate cluster was dedicated to fulfil this need. A straightforward and simple assumption can be to look for the closest public transportation stops to this k-medoid solution. As seen in Fig 2, this reduces the solutions to the outermost public transportation stops but does not solve the issue of occasional and unique rides. However, initializing a PFCMD clustering solution from the result of the k-medoid clustering will let these unique rides and look for the hubs containing

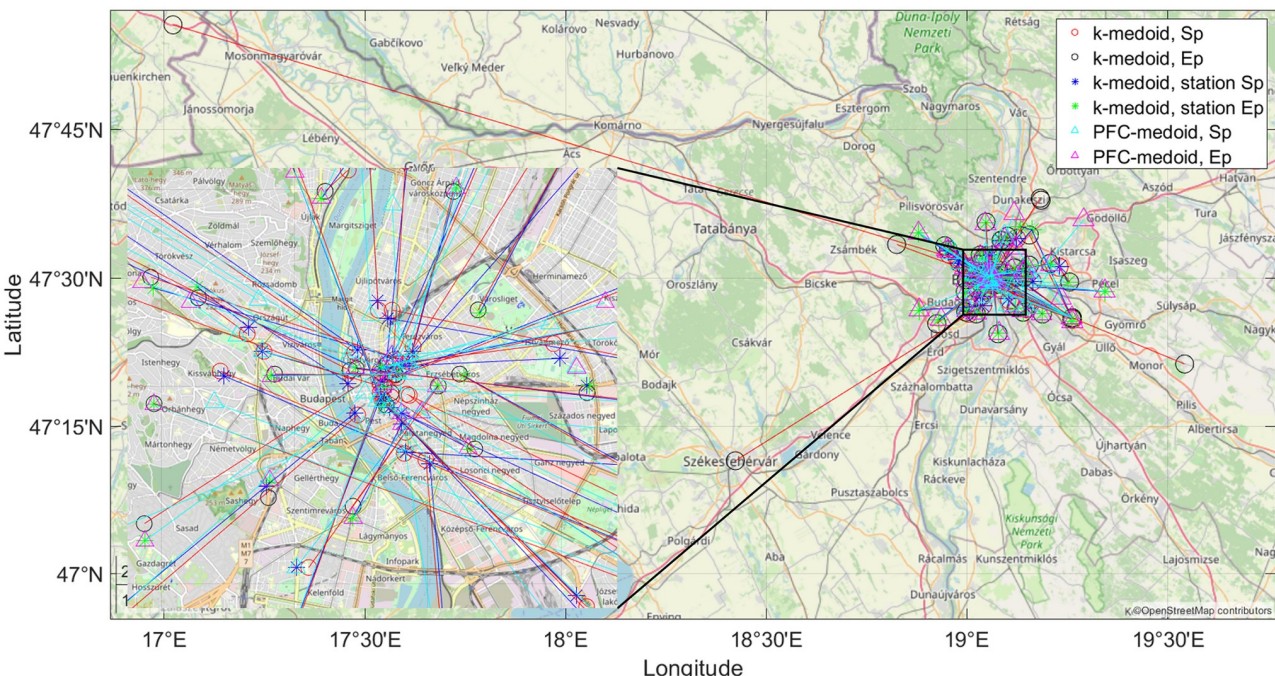

**Fig 2. The start- and end-points (Sp and Ep, respectively) of the k-medoid clustering of the Taxicab rides are marked with red and black circles, respectively.** The nearest public transportation stops to these start- and end-points are marked by black (Sp) and green (Ep) asterisks. In contrast, the clusters designed by the PFCMD algorithm are marked with cyan (Sp) and magenta (Ep) triangles, respectively. The colour of the line connecting the related stations in a straight line is the same as the colour of the starting station. The problem of outliers is well-reflected in the case of the trip to Vienna (red line on the left side of the figure) and constraining the solution to the outermost public transportation stop does not solve the problem neither.

enough rides in a walkable distance. This algorithm is not just highly flexible, where the cluster centroids are selected not from the rides but the public transportation stops. However, the parameters allow a highly flexible setting that can be tailored for the requirements.

## Tuning the parameters of the clustering algorithm

The aim of this section is to present how these parameters can be fine-tuned to tailor the algorithm to meet the requirements of the analysis.

*The value of fuzziness exponent*, $m$: in the case of a crisp $m$ value (closer to one), the resultant clusters are going to be crisp as well, with no fuzziness introduced to the system. However, by increasing the fuzziness parameters, the borders of different clusters become more overlapping and less crisp. Choosing a too high fuzziness parameter is disadvantageous as well: as the membership values $u_{ik}$ are less than one, taking them on a high $m$ exponent results in a minimal number. Therefore, the cluster members are primarily determined by the typicality values $\tau_{ik}$. This effect of parameter $m$ is discussed in depth with detailed experiments in Pal *et al.* [38]. For specific datasets, this parameter can be tuned based on the effects of outliers: starting with one, crisp clusters are generated, while increasing its value, the effect of outliers is reduced. The optimal value is tuned experimentally, typically between one and two; however, higher values are possible as well. In the present work, to avoid highly amorphous clusters, a relatively strict $m$ parameter was chosen as 1.2.

*The parameters of typicality*, $\eta, \alpha, \beta, \gamma$: As it was formerly described in the Method section, the original typicality function introduced by Pal *et al.* [38] was replaced by a function showing a decreasing trend as presented on Fig 3. The shape of this function aims to symbolize the

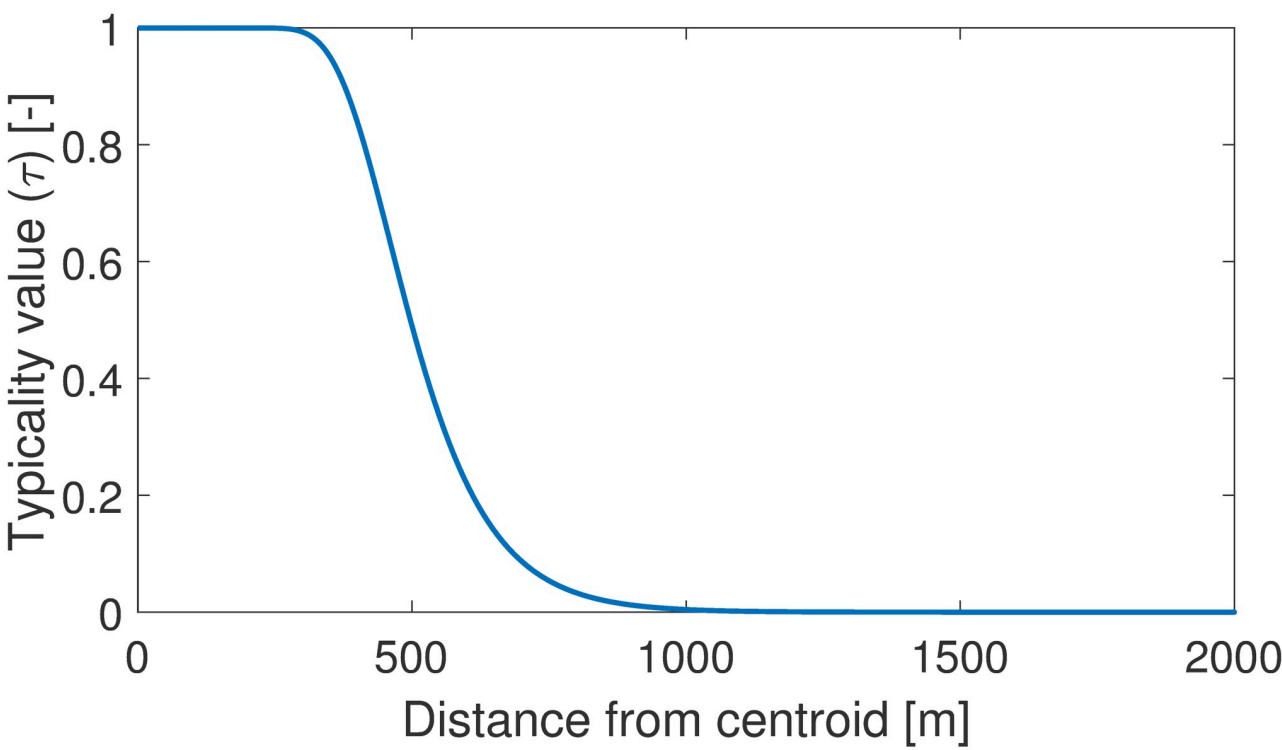

**Fig 3. The typicality function represents the willingness of passengers to walk to a nearby public transportation stop.** This function implicitly determines how many clusters are needed to group rides nearby the transportation lines.

willingness of a passenger to walk between the origin or destination of his/her travel and the nearest public transportation stop. The parameters are chosen to represent an approximately $500m$ range in which the passengers happily walk, but between 500 and $1000m$ this willingness rapidly drops ($\alpha = 1, \beta = 100, \gamma = 0.01$). This distance calculation is performed by considering the L2 norm distance of the start- and end-point of the travel and the public transportation stops. To preserve the shape of the typicality function and, thus, its physical meaning, the $\eta$ is chosen as 1.

*The coefficients of the membership function*, $a$ and $b$: the choice of these coefficients or weights describes the emphasis on the membership and typicality values. In order to reduce the effect of outliers, the value of $b$ is to be increased. However, by an increased value of $a$, the effect of membership values is favoured. In the present context, the typicality part of the equation constraints the collection of rides starting and/or ending far away from each other into the same cluster. Therefore, as in the present work, public transportation lines are to be designed, where the walking distance to and from the stops is a crucially important aspect of applicability. We put a much higher emphasis on the typicality values and chose $a = 0.1$ and $b = 0.9$.

*The number of clusters*, $C$: The parameter $C$ defines the number of cluster centroids. The final number of public transportation lines can be different: the routes in similar directions can be merged, or different clusters can have the same public transportation stops as their centroids. This provides the opportunity to find the dense hubs and serve their needs in public transportation service. By setting a relatively high parameter $C$ allows flexibility to the algorithm to optimally populate the clusters and hence, we can select the truly significant ones (the ones containing a significantly high number of rides in the clusters.) The true number of new

public transportation lines can be determined after analysing the resultant clusters and their comparison to the existing lines. According to this consideration we selected the number of clusters to cover a wide range of travels and applied two cluster validity measures to validate appropriateness of the number of the clusters. Finally, we set the $C$ parameter to be 50. The 0.9296 PC partition coefficient and the 0.4322 CE classification entropy indicate that the algorithm generated well-separated partitions with the selected settings.

## Spatial analysis of the resultant clusters

Fig 4 shows the number of rides in each cluster (bar plot) and the proportion of covered rides on the line plot. Evidently, most of the clusters consist of a few rides, but this and the following analysis and visualization results underpin our assumption that these rides form a very sparse system in which we need to determine the most significant hubs. Consequently, only a small fraction, less than 3% of the rides, can be covered by bus lines using these strict constraints on the walking distances. Fig 4 also shows that by selecting a higher number of clusters, parameter $C$ of the PFCMD algorithm, we provide flexibility to the algorithm so that it can populate the available clusters. After the clustering step, the clusters with insufficient number of rides in them can be neglected.

The designed lines containing at least one ride are visible together in Fig 5 (the algorithm places as many clusters as desired, even if no rides fulfil the required criteria). Naturally, these clusters can be further filtered based on the more in-depth aspects of the public transportation experts.

Fig 6 showcases some examples of the resultant clusters of the PFCMD algorithm. The start- and end-points points are represented by yellow dots and black crosses, respectively. The

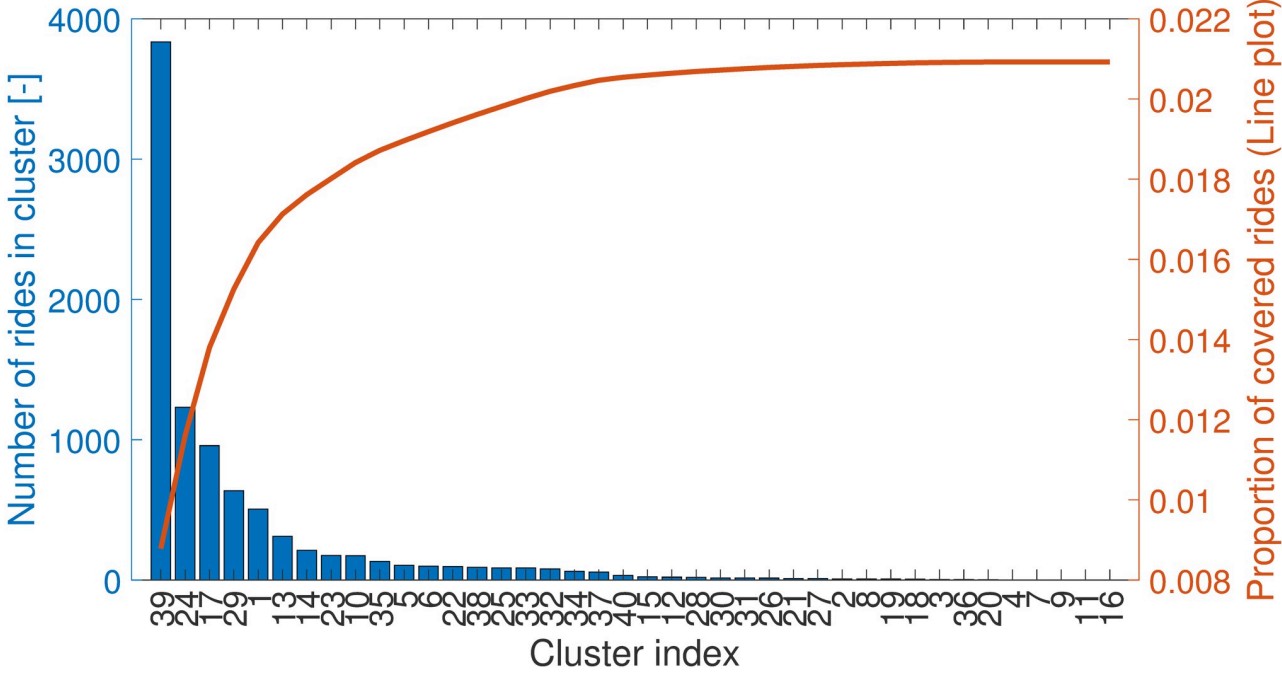

**Fig 4. The number of data points in each cluster if the threshold of membership is $P_{thres}$ = 0.15 (bar plots, left axis) and the proportion of rides covered by the clusters compared to all the analysed Taxicab rides during the nights (line plot, right axis).** Some clusters have low importance due to the few supporting passengers (bar plots in part (b)). As the Taxicab rides usually handle unique and occasional travels, a small percentage of the rides can be replaced by public transportation lines.

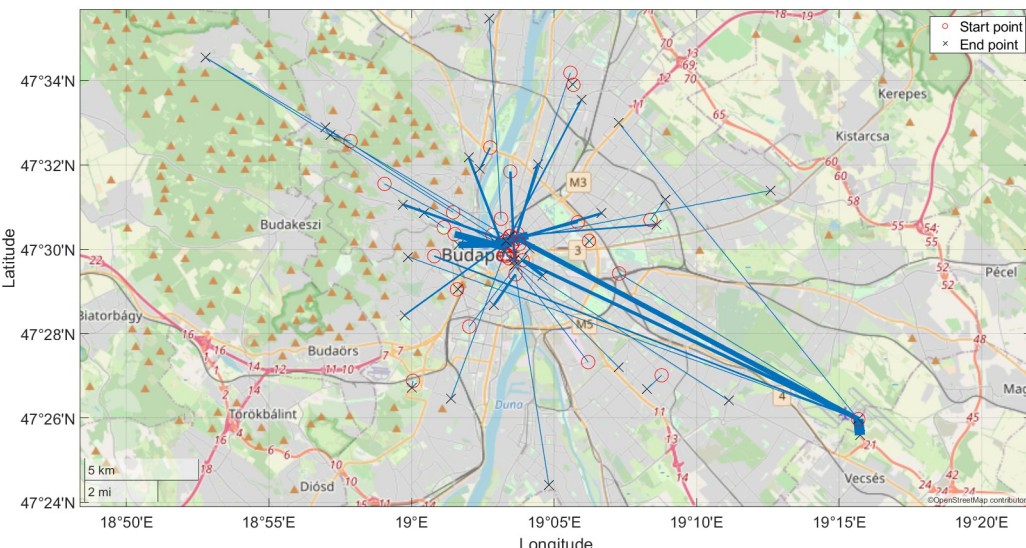

**Fig 5. The cluster centroids are represented as lines on the map of Budapest.** The start and end points of a recommended public transport line are marked by yellow circles and black crosses, respectively. The width of the line is proportional to the size (and hence, importance) of the cluster. The narrower lines represent smaller, while the wider lines represent bigger clusters.

arrow at each sub-figure connects the public transportation stops serving as the start- and end-points for the designed line, and points in the travelling direction. Every centre of the cluster is a public transportation stop in Budapest. Evidently, one of the most important hubs is the Budapest Ferenc Liszt International Airport, as clusters 8 and 25 (part (a) and (c) of Fig 6) point there and clusters 17 (part (b) of Fig 6), 31 and 45 origin from the international airport. Moreover, cluster 36 is responsible for shorter rides near the airport. As expected, the other dense area with numerous clusters pointing into and from is the inner city centre, where most of the events occur at nighttime. Clusters 26, 32 and 48 (part (d), (e) and (f)) are good examples of this.

### Temporal analysis of the resultant clusters

The presented analysis not only calls attention to missing transportation directions but also recommends the schedule of these lines. After identifying the cluster members, the time schedule of the lines is analysed as well.

The proposed approach assumes that there are typical time periods (*e.g.*, typical Monday mornings) that can be aggregated for the analysis. These periods were determined by the exploratory data analysis of the number of travels. Fig 7 shows day-wise and hour-wise box-plots of the distribution number of the rides. These boxplots illustrate the number of rides at the specific temporal period. The data points can be grouped at seasonal intervals to reveal how the values are distributed within the days of the week and the hours of the day, and how this compares over time shows the day-wise breakdown of the Taxicab. The busiest days are Saturday and Friday when many people are most likely to arrive at the city for entertainment during the night and book Taxicabs to the city center. Similar plots can be generated for the hour-wise (or any temporal resolution) breakdown of the rides in a cluster.

By analyzing the start times of the rides within all clusters, suggestions can be made for the schedule of the proposed lines. Fig 8 shows the time-series analyses of all clustered rides. Based

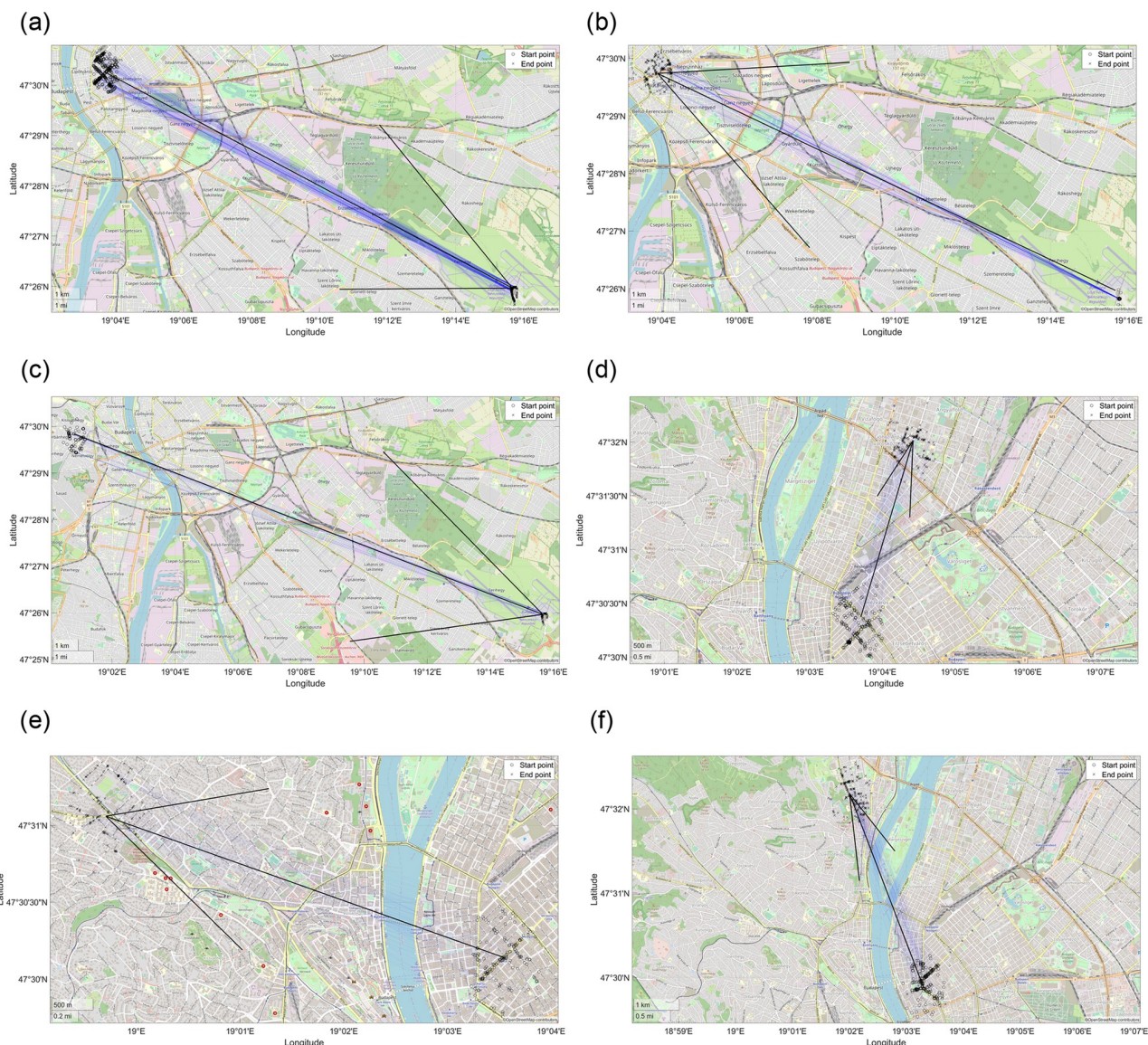

**Fig 6. Exemplary clusters, where the start and endpoints points of the individual rides in the cluster are represented by yellow dots and black crosses, respectively.** The cluster centroids, marked by the start and end point of the arrow, point from one public transport stop to another.

on the analysis of the start time of the Taxicab rides, we can notice that during the weekdays, the distributions of the Taxicab usages are the same. The busy periods during the night can be determined: these are the optimal periods when the related line is most likely to take advantage.

The proposed analysis can be performed with any temporal resolution of interest. A detailed overview of the temporal analysis solutions of Taxicab data and the determination of busy periods was presented by Varga *et al.* [16]. For example, by using a sufficiently fine temporal resolution (e.g., 60, 30, or 15-minute windows) and counting the number of rides starting in the specific window, the public transportation line can be scheduled for the busy periods where the lines are most needed.

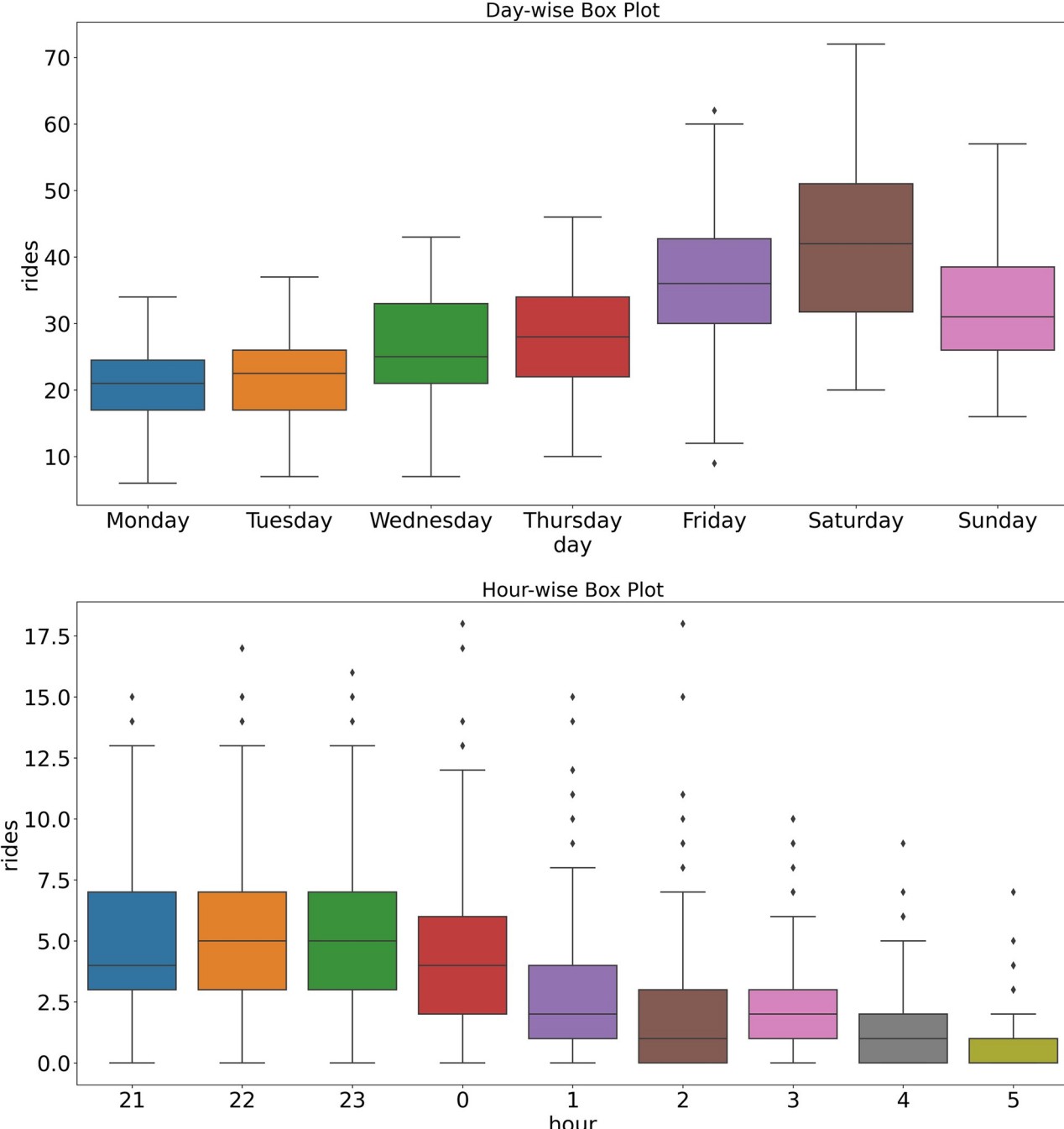

**Fig 7. Temporal analysis of night rides on the full dataset.** The trend shows an increase during the week until the Saturdays. The hour-wise analysis shows a constant usage before midnight and a decreasing trend before 5 am. Also, we can notice the outliers on the hour-wise boxplot. These are coming from the different characteristics of the weekends.

## Discussion

It is well known that Taxicab rides reasonably well represent human mobility patterns [17]. As the daytime public transportation system of Budapest is relatively dense and transparent with sometimes multiple parallel opportunities, in the present work we concentrated on Taxicab

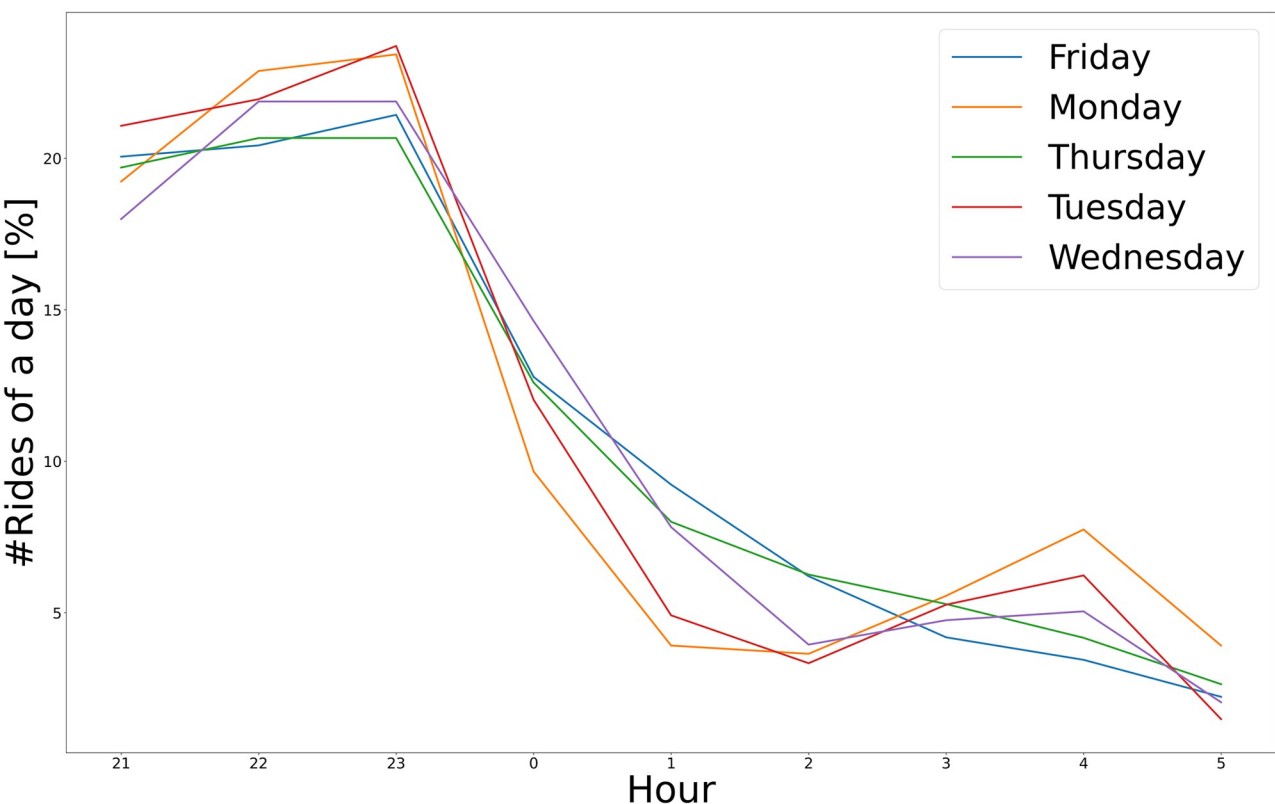

**Fig 8. Time-series analysis of the clustered dataset.** The characteristics of the rides are close to each other during the weekdays.

rides occurring at nights as in these cases, the venues of the different events (start and end of theatre plays or cinema movies, parties) are covered by the transport system less thoughtfully and purposefully. The resultant clusters show a solid connection to the nightlife of Budapest and people travelling to or from the airport. As we can see, the individual Taxicab rides at nighttime provide a relatively sparse coverage of the city, making it difficult to reasonably connect the rides and reduce the effect of outliers. The simple k-medoid-based algorithms may tend to incorporate outliers in the clusters, resulting in a very high bias of the data model. However, the proposed PFCMD overcomes the problem of outliers. Moreover, the goal-oriented typicality function supports incorporating user-defined desirability functions based on the walking distance to and from the public transportation stops. The designed clusters highlight two frequent areas of the nightlife of the city: the centre, with its numerous entertainment and recreation opportunities, and the airport, where the planes frequently take off and land in the very late and early hours. The temporal analysis of the clustered rides supports a more sustainable planning of the public transportation lines' timetable. However, it is evident that after the clustering of the analysed dataset, there are not enough rides in some clusters to further analyse the dynamics of the rides. This dataset can be considered as a sampling of the Taxicab rides available in Budapest, as a single Taxicab company provides the data. However, the mobility pattern is well-reflected in the results: in the data recording (2014), there was no direct line to the airport in Hungary, which was implemented in 2017 (with line ID 100E).

The results illustrate that the method is suitable to call attention to missing transportation lines and recommends the scheduling of these lines. However, it has to be noted that the clusters do not directly represent optimized routes; the clustering algorithm generates suggestions

for the experts by summarizing the demands in a sophisticated and robust way. Moreover, the derived areas and departure times provide precious information for Taxicab drivers. These are the potential places where they can more easily secure a ride in the related time slots.

## Conclusions

In the present work, the importance of human mobility patterns-based public transportation design in sustainable cities is discussed. A new clustering algorithm is developed to assign the GPS based patterns to pre-defined centre points. The proposed possibilistic fuzzy c-medoid (PFCMD) clustering algorithm can group the human mobility patterns to the existing public transportation stops places within a walkable distance. Based on the analysis of the resultant clusters, further insights into the dynamics of the city can be derived. The applicability of PFCMD is presented on the analysis of the GPS data of Taxicabs, assigning them to the public transportation stop place coordinates in the city of Budapest, Hungary. The results show some potential routes where the re-scheduled public transportation (buses), can replace Taxicab rides during the night shift. The temporal analysis of the clustered rides shows the potential days and times of the day to re-design the lines. To stimulate further research, the resultant MATLAB codes for the proposed possibilistic fuzzy c-medoid (PFCMD) clustering algorithm, is publicly available on the website of the authors (www.abonyilab.com).

## Author Contributions

**Conceptualization:** Miklós Mezei, György Eigner, Gyula Dörgő, János Abonyi.

**Data curation:** Gyula Dörgő, Tamás Ruppert.

**Formal analysis:** Tamás Ruppert.

**Funding acquisition:** György Eigner.

**Investigation:** Gyula Dörgő, Tamás Ruppert.

**Methodology:** Gyula Dörgő.

**Project administration:** György Eigner, János Abonyi.

**Resources:** Miklós Mezei, György Eigner.

**Software:** Miklós Mezei, Gyula Dörgő, Tamás Ruppert.

**Supervision:** Imre Felde, György Eigner, János Abonyi.

**Validation:** Imre Felde, Tamás Ruppert, János Abonyi.

**Visualization:** Gyula Dörgő, Tamás Ruppert.

**Writing – original draft:** Miklós Mezei, Gyula Dörgő, Tamás Ruppert.

**Writing – review & editing:** Imre Felde, György Eigner, János Abonyi.

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
