## [Decision Letter · Decision Letter 0]

14 Apr 2022

PONE-D-22-03021Goal-oriented possibilistic fuzzy C-Medoid clustering of human mobility patterns – Illustrative application for the Taxicab trips-based enrichment of public transport servicesPLOS ONE

Dear Dr. Eigner,

Thank you for submitting your manuscript to PLOS ONE. After careful consideration, we feel that it has merit but does not fully meet PLOS ONE’s publication criteria as it currently stands. Therefore, we invite you to submit a revised version of the manuscript that addresses the points raised during the review process.

We look forward to receiving your revised manuscript.

Kind regards,

Luca Pappalardo

Academic Editor

PLOS ONE

Journal Requirements:

2. We note that Figure 1, 3, 4, 5a, 5b, 5c, 5d, 5e and 5f in your submission contain [map/satellite] images which may be copyrighted. All PLOS content is published under the Creative Commons Attribution License (CC BY 4.0), which means that the manuscript, images, and Supporting Information files will be freely available online, and any third party is permitted to access, download, copy, distribute, and use these materials in any way, even commercially, with proper attribution. For these reasons, we cannot publish previously copyrighted maps or satellite images created using proprietary data, such as Google software (Google Maps, Street View, and Earth). For more information, see our copyright guidelines: http://journals.plos.org/plosone/s/licenses-and-copyright.

   a. You may seek permission from the original copyright holder of Figure 1, 3, 4, 5a, 5b, 5c, 5d, 5e and 5f to publish the content specifically under the CC BY 4.0 license. 

“NO authors have competing interests.”

“Project no. 2019-1.3.1-KK-2019-00007. has been implemented with the support provided from the National Research, Development and Innovation Fund of Hungary, financed under the 2019-1.3.1-KK funding scheme.

Funder: National Research, Development and Innovation Office

https://nkfih.gov.hu/for-the-applicants

The funders did not play any role regarding the study.”

Additional Editor Comments (if provided):

The work is potentially interesting but it requires significant improvements.

Reviewers' comments:

Reviewer's Responses to Questions

**Comments to the Author**

1. Is the manuscript technically sound, and do the data support the conclusions?

Reviewer #1: Partly

Reviewer #2: Partly

2. Has the statistical analysis been performed appropriately and rigorously? 

Reviewer #1: Yes

Reviewer #2: N/A

3. Have the authors made all data underlying the findings in their manuscript fully available?

Reviewer #1: No

Reviewer #2: No

4. Is the manuscript presented in an intelligible fashion and written in standard English?

Reviewer #1: No

Reviewer #2: Yes

5. Review Comments to the Author

Reviewer #1: The idea and scope of the paper are interesting. In particular, the application of such a clustering algorithm with a tipicality function used to model people's willingness to walk to a bus stop is interesting.

However, the work is not mature enough. Both the introduction and results sections are often unclear, the experiments performed are quite poor, and the figures provided do not help at all at understanding.

In general, I think this work has potential, but it is not ready to be published.

Below specific comments:

- Introduction

Sometimes very confused. It is often not clear whether the authors are referring to something that has been done in another work or they are stating something not related to any another work. I would suggest to use the formula "Authors et al. [X] studied/analysed/stated/ ..." to let the reader understand that you are referring to someone else's work.

rows 4-6: "Based on these facts, innovative changes are needed to mitigate the effects of climate change within cities, and the transport sector there is one of the main contributors to greenhouse gas emissions"

this is not very clear, may be rephrased.

rows 31-33: "These GPS trajectories could be used to analyse the emission of particle matter from braking behaviours"

This statement is a bit out of context here. I would move it between the examples on how GPS trajectories can be used (from row 36 on).

row 47: I would add "In particular, electric buses [...]".

row 78-79: avoiding the ad-hoc installation of new stops is more of a choice than a major deficiency of clustering application for the design of public transportation lines. I would cite this as a decision made in the authors' paper, that has both a practical and a methodological motivation, but not as a deficiency of these particular clustering applications.

rows 110-113: "The developed PFCMD clustering algorithm that aims to cluster [...]" note clear, maybe a typo (remove "that"?)

- Materials and Methods

This section is very well written and clear.

eqs. (4) and (5): what does it mean "inA" in the end of the equation?

- Figures

In general, the figures are unintelligible: they have very low resolution, too small markers in the legend, it is almost impossible to even get the colors of the markers.

Fig. 1: colors in the legend (especially in the plot on the right) are unintelligible

Fig. 2: it seems it is missing the part of this figure referenced at row 297

Fig. 3: a zoom on the right hand side of the figure would definitely help. The figure as it stands is only useful for getting the outlier theme, but not all the rest.

- Results

rows 261-268: any reference / detail / experiment about the choice of m?

- Discussion

rows 343-344: "In our work, we demonstrate that Taxicab rides represent the human mobility patterns reasonably well (similarly to [18])." this does not seem the goal of the authors' work nor what they have done.

- References

Refs [14] and [21] refer to the same work

Refs [41] and [45] refer to the same work

Reviewer #2: In this paper, the authors propose a possibilistic fuzzy c-medoid clustering algorithm to study human mobility. The proposed medoid-based clustering approach groups the typical mobility patterns within walking distance to the stations of the public transportation system. Their results demonstrate that the mobility pattern is well-reflected. In fact, in the data recording (2014), there was no direct line to the airport in Hungary, which was implemented later in 2017.

The manuscript and the analysis have the potential to improve the knowledge in the human mobility field; however, I have some concerns about the methodology and some improvements that can be applied to increase the readability and the quality of the proposed manuscript:

1. The sentence starting in row 110 "The developed PFCMD clustering algorithm that aims to cluster.." seems to have an unnecessary “that”;

2. The introduction is sometimes not flowy, and it is not easy to understand what the references of some previous works refer to;

3. The time series analysis is not deep enough. I expect a deeper discussion about this crucial aspect for this work;

4. The clustering is performed for the taxi rides collected for a whole year, and after that, some public transport lines are suggested from the result of the clustering. I believe that the fact that the lines are indicated over an entire year's data and their influence on the study should be discussed better;

5. In line 156 I would say “the clustering is performed in the geographical domain” instead of “the clustering is performed in the spatial space”;

6. What is the “inA” at the end of equations 4 and 5? it is not clear to me;

7. The section “Temporal analysis of the resultant clusters” should be extended and described better;

8. In the sub-section “The number of public transportation lines, c:” the fact that the number of clusters may affect the number of lines is repeated and this may appear redundant. Furthermore, I suggest selecting c in such a way that optimizes a designed objective function;

9. How was the parameter m selected? it should be discussed;

10. How is the clustering quality measured?

11. In line 311 and 340 the authors stated “as seen/illustrated in Figure, …” the number of the Figure should be provided;

12. I think it is necessary to define an algorithm that performs the suggestions that can be made for the schedule of the lines. In general the rationale behind those choices are not that clear to me;

13. The quality of the Figures should be imporved to help the reader understand better some concepts of this manuscript, for example, they have a very low resolution and the markers in the legend are too small.

6. PLOS authors have the option to publish the peer review history of their article (what does this mean?). If published, this will include your full peer review and any attached files.

Reviewer #1: **Yes: **Matteo Bohm

Reviewer #2: No

---

## [Author Response · Author response to Decision Letter 0]

25 Jun 2022

Reply to reviewers

Title: Goal-oriented possibilistic fuzzy C-Medoid clustering of human mobility patterns – Illustrative application for the Taxicab trips-based enrichment of public transport services

Authors: Miklós Mezei, Imre Felde, György Eigner, Gyula Dörgő, Tamás Ruppert, János Abonyi

Ref. no.: PONE-D-22-03021

Dear Editor, 

We are grateful for the useful and supporting comments of the reviewers. We have addressed all the suggestions as explained below. We would like to thank the reviewers for their constructive comments. The changes in the manuscript are highlighted in blue. We hope that the revised manuscript meets your expectations.

About the copyrights of our figures. The research group obtains an academic Matlab license, which is legally suitable for publishing the results generated with its use. All figures are created by geoplot (https://www.mathworks.com/help/matlab/ref/geoplot.html) and geoscatter (https://www.mathworks.com/help/matlab/ref/geoscatter.html) Matlab toolboxes, which are commonly used for visualization, and the reslulted plots can be inserted into open source documents. We hope these are fit to the journal copyright policy.

Sincerely yours, 

György Eigner

Reviewer 1

The idea and scope of the paper are interesting. In particular, the application of such a clustering algorithm with a tipicality function used to model people's willingness to walk to a bus stop is interesting.

However, the work is not mature enough. Both the introduction and results sections are often unclear, the experiments performed are quite poor, and the figures provided do not help at all at understanding.

In general, I think this work has potential, but it is not ready to be published.

Below specific comments:

- Introduction

Sometimes very confused. It is often not clear whether the authors are referring to something that has been done in another work or they are stating something not related to any another work. I would suggest to use the formula "Authors et al. [X] studied/analysed/stated/ ..." to let the reader understand that you are referring to someone else's work.

Thank you very much for the suggestion. After the careful revision of the introduction, we agree that the flow of the text needed a significant improvement. We added phrases to clarify that we refer to the work of other researchers and build on their findings. We hope that the readability and flow of the text have increased significantly.

rows 4-6: "Based on these facts, innovative changes are needed to mitigate the effects of climate change within cities, and the transport sector there is one of the main contributors to greenhouse gas emissions"

this is not very clear, may be rephrased.

Thank you very much. The cited sentence was indeed badly phrased. We have rephrased it: “The transport sector is one of the main contributors to greenhouse gas emission. Rapid urban population growth, traffic congestion, and related air pollution put cities at the center of the climate mitigation agenda. These facts suggest urgent and transformative actions in urban mobility are required [2].” Also, we read the manuscript carefully and corrected the grammar and misspellings.

rows 31-33: "These GPS trajectories could be used to analyse the emission of particle matter from braking behaviours"

This statement is a bit out of context here. I would move it between the examples on how GPS trajectories can be used (from row 36 on).

Thank you very much for the valuable suggestion. We have moved it to the suggested part of the manuscript.

row 47: I would add "In particular, electric buses [...]".

Thank you very much. We have added the recommended phrase.

row 78-79: avoiding the ad-hoc installation of new stops is more of a choice than a major deficiency of clustering application for the design of public transportation lines. I would cite this as a decision made in the authors' paper, that has both a practical and a methodological motivation, but not as a deficiency of these particular clustering applications.

Thank you very much for your carefulness. We have corrected the title of the list, stating that here we list the three significant deficiencies and practical problems/aspects.

rows 110-113: "The developed PFCMD clustering algorithm that aims to cluster [...]" note clear, maybe a typo (remove "that"?)

Thank you very much for your carefulness. We have removed the unnecessary “that” from the sentence.

- Materials and Methods

This section is very well written and clear.

eqs. (4) and (5): what does it mean "inA" in the end of the equation?

Thank you very much. We have added a more detailed description to the paragraph next to the equations on page 6 and removed the unnecessary DinA notation that was introduced to prepresent a general distance norm.

- Figures

In general, the figures are unintelligible: they have very low resolution, too small markers in the legend, it is almost impossible to even get the colors of the markers.

Fig. 1: colors in the legend (especially in the plot on the right) are unintelligible

Thank you very much for this crucial highlight. We increased the marker on the legend.

Fig. 2: it seems it is missing the part of this figure referenced at row 297

Thank you, you highlighted that we need to separate these two figures. Now you can find these in Figures 3 and 4 with the references at the end of “Tuning the parameters of the clustering algorithm” (Fig. 3) and the beginning of the “Spatial analysis of the resultant clusters” (Fig 4) sections.

Fig. 3: a zoom on the right hand side of the figure would definitely help. The figure as it stands is only useful for getting the outlier theme, but not all the rest.

The figure aims to show the outliers as the nature of the compared clustering algorithms. We modified the figure, now it is Fig. 2. We created a subplot inside the original one, where we zoom in to the city center to show the rest of the cluster parallel with the outliers.

- Results

rows 261-268: any reference / detail / experiment about the choice of m?

Thank you very much for the insightful question. We have extended the description of the effect and choice of the parameter m and also referenced the work of Pal et al., where detailed experiments are present. “[...] effect of parameter $m$ is discussed in-depth with detailed experiments in Pal et al. [38]. For specific datasets, this parameter can be tuned based on the effects of outliers: starting with 1, crisp clusters are generated, while increasing its value, the effect of outliers is reduced. The optimal value is tuned experimentally, typically between 1 and 2. However, higher values are possible as well.”

- Discussion

rows 343-344: "In our work, we demonstrate that Taxicab rides represent the human mobility patterns reasonably well (similarly to [18])." this does not seem the goal of the authors' work nor what they have done.

Thank you very much for the suggestion, we agree with that. We have rephrased the relevant part: “In our work, we build on the assumption that Taxicab rides represent the human mobility patterns reasonably well...” Also, your comment highlighted us we need to make a clear contribution. Now you can find it at the end of the Discussion section.

- References

Refs [14] and [21] refer to the same work

Refs [41] and [45] refer to the same work

Thank you very much for your carefulness. We have corrected the references. Thank you for all the constructive critics and careful revision. We truly believe that based on these improvements, the quality of the manuscript has increased significantly.

Reviewer 2

In this paper, the authors propose a possibilistic fuzzy c-medoid clustering algorithm to study human mobility. The proposed medoid-based clustering approach groups the typical mobility patterns within walking distance to the stations of the public transportation system. Their results demonstrate that the mobility pattern is well-reflected. In fact, in the data recording (2014), there was no direct line to the airport in Hungary, which was implemented later in 2017.

The manuscript and the analysis have the potential to improve the knowledge in the human mobility field; however, I have some concerns about the methodology and some improvements that can be applied to increase the readability and the quality of the proposed manuscript:

1. The sentence starting in row 110 "The developed PFCMD clustering algorithm that aims to cluster.." seems to have an unnecessary “that”;

Thank you very much for your carefulness. We have removed the unnecessary “that” from the sentence.

2. The introduction is sometimes not flowy, and it is not easy to understand what the references of some previous works refer to;

Thank you very much for the critical criticism. After the careful revision of the introduction, we agree that the flow of the text needed a significant improvement. We reconstructed the sentences aiming to clarify the referred works. We hope that the readability and flow of the text have increased significantly.

3. The time series analysis is not deep enough. I expect a deeper discussion about this crucial aspect for this work;

AND

4. The clustering is performed for the taxi rides collected for a whole year, and after that, some public transport lines are suggested from the result of the clustering. I believe that the fact that the lines are indicated over an entire year's data and their influence on the study should be discussed better;

We appropriate your comments. These are helpful and improved our manuscript a lot. We did not focus enough to prove our hypothesis that there are typical patterns during the rides in the time series. We extended the temporal analysis and the analysed taxicab rides sections. We added new plots to show the characteristic of the number of rides during the night shift. In Figure 8, you can notice the similar characteristic of the weekdays. We regenerated the boxplot analyses based on the complete datasets considering the 436 537 rides during the entire year (at night, between 9 p-m and 5 a.m). Figure 7 shows the new results. The additional information confirms that the proposed clustering algorithm explores relevant cluster in the time domain.

5. In line 156 I would say “the clustering is performed in the geographical domain” instead of “the clustering is performed in the spatial space”;

Thank you very much for your carefulness. It is corrected.

6. What is the “inA” at the end of equations 4 and 5? it is not clear to me;

Thank you very much. We have added a more detailed description to the paragraph next to the equations on page 6 and removed the unnecessary DinA notation that was introduced to prepresent a general distance norm.

7. The section “Temporal analysis of the resultant clusters” should be extended and described better;

AND

12. I think it is necessary to define an algorithm that performs the suggestions that can be made for the schedule of the lines. In general the rationale behind those choices are not that clear to me;

Thank you very much for the crucial comment. Revisiting the section, we agree that this part of the method was not detailed enough. We extended the description of the section, describing how the analysis of the start time of the rides in a specific cluster supports the determination of optimal public transportation schedules. Moreover, we highlighted that the work of Varga et al. describes in detail the value and method of temporal analysis of taxi rides. The discussion of the paper has been also extended to discuss that the algorithm generates suggestions for the experts by summarizing the demands in a sophisticated way. . 

8. In the sub-section “The number of public transportation lines, c:” the fact that the number of clusters may affect the number of lines is repeated and this may appear redundant. Furthermore, I suggest selecting c in such a way that optimizes a designed objective function;

Thank you very much for the important remark. We have chosen the parameter c, the number of clusters to be sufficiently high and allow flexibility for the algorithm. This way, the algorithm can populate these clusters with the rides, and we can analyse the resultant clusters considering only the clusters with a sufficient number of rides. We do not constrain the algorithm by strictly defining the number of clusters in advance. As shown in Fig. 4., this way, only a well-defined number of clusters will be statistically interesting for us based on the number of incorporated rides in the clusters. 

Thank you again for this important suggestion. We have extended the text with this further description. We hope this approach will clarify our parameter selection approach to the reader.

9. How was the parameter m selected? it should be discussed;

Thank you very much for the insightful question. We have extended the description of the effect and choice of the parameter m and also referenced the work of Pal et al., where detailed experiments are present. “[...] effect of parameter $m$ is discussed in-depth with detailed experiments in Pal et al. For specific datasets, this parameter can be tuned based on the effects of outliers: starting with one, crisp clusters are generated, while increasing its value, the effect of outliers is reduced. The optimal value is tuned experimentally, typically between one and two. However, higher values are possible as well.”

10. How is the clustering quality measured?

Thank you for this crucial comment. We can measure the quality of the cluster in two ways. Firstly, we can interpret the results (as we did in the “Clustering of Public Transportation Data” section). Secondly, we can apply cluster validity measures. For that, we extended the Methods section with two equations (eq. 7 and 8) to define the Partition Coefficient (PC) and Classification Entropy (CE). These are complex measures, thanks to that, these are independent of the distance values. The high PC and low CE measures both claim a well-separated cluster structure, which underpin our results.

11. In line 311 and 340 the authors stated “as seen/illustrated in Figure, …” the number of the Figure should be provided;

Thank you very much, we have added the number of the figures to the description. (The part at line 340 has changed, but the exact number of the particular Figure is provided in every case.)

13. The quality of the Figures should be imporved to help the reader understand better some concepts of this manuscript, for example, they have a very low resolution and the markers in the legend are too small.

Thank you very much for this crucial highlight. We increased the marker on the legend of Figure 2 (the previous version of the manuscript was Figure 1). We separated the typicality and the number of data points figures into individuals. Now you can find these in Figures 3 and 4. Also, we create a subplot inside the original Figure 2 (the previous version of the manuscript was Figure 3). We zoom in to the city center to show the rest of the cluster parallel with the outliers. We replaced the figure of the boxplots (Fig. 7) and put it in the temporal analyses to prove the datasets. Also, we created a new plot to prove the characteristics of the rides. You can see it in Figure 8.

---

## [Decision Letter · Decision Letter 1]

6 Sep 2022

Goal-oriented possibilistic fuzzy C-Medoid clustering of human mobility patterns – Illustrative application for the Taxicab trips-based enrichment of public transport services

PONE-D-22-03021R1

Dear Dr. Eigner,

We’re pleased to inform you that your manuscript has been judged scientifically suitable for publication and will be formally accepted for publication once it meets all outstanding technical requirements.

Kind regards,

Yajie Zou

Academic Editor

PLOS ONE

Additional Editor Comments (optional):

Reviewers' comments:

Reviewer's Responses to Questions

**Comments to the Author**

1. If the authors have adequately addressed your comments raised in a previous round of review and you feel that this manuscript is now acceptable for publication, you may indicate that here to bypass the “Comments to the Author” section, enter your conflict of interest statement in the “Confidential to Editor” section, and submit your "Accept" recommendation.

Reviewer #1: All comments have been addressed

Reviewer #3: All comments have been addressed

2. Is the manuscript technically sound, and do the data support the conclusions?

Reviewer #1: (No Response)

Reviewer #3: Yes

3. Has the statistical analysis been performed appropriately and rigorously? 

Reviewer #1: (No Response)

Reviewer #3: Yes

4. Have the authors made all data underlying the findings in their manuscript fully available?

Reviewer #1: (No Response)

Reviewer #3: Yes

5. Is the manuscript presented in an intelligible fashion and written in standard English?

Reviewer #1: (No Response)

Reviewer #3: Yes

6. Review Comments to the Author

Reviewer #1: (No Response)

Reviewer #3: (No Response)

7. PLOS authors have the option to publish the peer review history of their article (what does this mean?). If published, this will include your full peer review and any attached files.

Reviewer #1: No

Reviewer #3: No

---

## [Editor Report · Acceptance letter]

27 Sep 2022

PONE-D-22-03021R1 

Goal-oriented possibilistic fuzzy C-Medoid clustering of human mobility patterns – Illustrative application for the Taxicab trips-based enrichment of public transport services 

Dear Dr. Eigner:

I'm pleased to inform you that your manuscript has been deemed suitable for publication in PLOS ONE. Congratulations! Your manuscript is now with our production department. 

Kind regards, 

on behalf of

Dr. Yajie Zou 

Academic Editor

PLOS ONE